# MINIMALISTIC PREDICTIONS FOR ONLINE CLASS CONSTRAINT SCHEDULING

**Dorian Guyot**
University of Fribourg
Fribourg, Switzerland
dorian.guyot@unifr.ch

**Alexandra Lassota**
Eindhoven University of Technology
Eindhoven, The Netherlands
a.a.lassota@tue.nl

## ABSTRACT

We consider online scheduling with class constraints. That is, we are given $m$ machines, each with $k$ class slots. Upon receiving a job $j$ with processing time $p_j$ and class $c_j$, an algorithm needs to allocate $j$ on some machine $i$. The goal is to minimize the makespan while not assigning more than $k$ different classes onto each machine. While the offline case is well understood and even (E)PTAS results are known [Chen Jansen Luo Zhang COCOA'16; Jansen, Lassota, Maack SPAA'20], the online case admits strong impossibility results in classical competitive analysis [Epstein, Lassota, Levin, Maack, Rohwedder STACS'22].

We overcome these daunting results by investigating the problem in a learning-augmented setting where an algorithm can access possibly erroneous predictions. We present new algorithms with competitive ratios independent of $m$ and tight lower bounds for several classical and problem-specific prediction models. We thereby give a structured overview of what additional information helps in the design of better scheduling algorithms.

## 1 INTRODUCTION

Makespan minimization on parallel identical machines is a fundamental and intensively studied problem and a classical example for online algorithm analysis through Graham's famous list scheduling algorithm Graham (1966). In this problem, jobs arrive online and upon arrival of a job $j$, an algorithm needs to assign $j$ to a machine. Note that there exist two main ways for the jobs to be made available to the algorithm: one by one or over time. We consider the former.

Motivated by its various applications in product planning, data placement, and load balancing, we consider a variant with class constraints Jansen et al. (2020). That is, we are given $m$ machines and $n$ jobs with job $j$ having processing time $p_j$ and class $c_j$. Each machine can schedule jobs from at most $k$ different classes. We say that a machine has $k$ *class slots*. A class slot on a machine $i$ is *occupied* by a class $c$ if at least one job of class $c$ is scheduled on $i$. We consider classical makespan minimization, that is, minimizing the maximum load on any machine:

$$\text{minimize} \quad \max_{i \in [m]} \sum_{j \text{ scheduled on } m_i} p_j \quad \text{s.t.} \quad \left| \{ c_j \mid j \text{ scheduled on } m_i \} \right| \leq k \quad \forall i \in [m].$$

A famous special case of this problem is cardinality constrained scheduling, where each class only contains a single job. Thus, the class restriction is equivalent to limiting the number of jobs on a machine. Cardinality constraints naturally appear in a variety of applications, ranging from distributing students in universities to assembly line optimizations or circuit board design Zhang et al. (2009).

The offline cases are well-understood. Both problems are known to be NP-hard Shachnai & Tamir (2000); Chen et al. (2016). This motivated an extensive study of approximation algorithms. Among the latest findings, there are PTAS results known for the class constraint scheduling variant relying on modeling the problem as well-structured integer programs, and even an EPTAS for the cardinality constraint special case Jansen et al. (2020); Chen et al. (2016).

For the online setting, no competitive ratio better than the trivial bound of $m$ can be achieved for class constraint scheduling. For cardinality constraint scheduling, a constant lower bound of 2 is known

Epstein et al. (2022). The competitive ratio $\max_{I \in \mathcal{I}} makespan(I)/makespan(OPT)$ is a widely used measurement for the performance of an online algorithm. It measures the maximum ratio over the set of all instances $\mathcal{I}$ between the online algorithm's ($I$) and an offline optimal solution's ($OPT$) objective value. Here, $OPT$ is the best schedule that can be computed with complete information and unbounded running time.

Although the lower bounds seem unreachable, the setting does not reflect the power of most real-world applications, where certain aspects of the input can be learned using historical data with application-sensitive modeling. We use this approach to investigate the problem in a learning-augmented setting where an algorithm can access such predictions (see, e.g., Mitzenmacher & Vassilvitskii (2022)). However, these predictions might not perfectly reflect the current instance. Measuring this error allows parameterized performance guarantees – most prominent among those are *consistency* and *robustness* – of algorithms w.r.t. the prediction model. Consistency is the competitive ratio for perfect predictions. Robustness is an upper bound on the competitive ratio for any prediction w.r.t. the error of this prediction.

The above definitions are generally applicable to all prediction models. Thus, another main ingredient is to select a reasonable and well-performing prediction model for the given problem. In this work, we study the two most prominent prediction models: *full input* predictions (see, e.g., Purohit et al. (2018); Azar et al. (2021); Im et al. (2021) and the references in the papers) and *action* predictions (see, e.g., Antoniadis et al. (2023); Bamas et al. (2020); Lindermayr & Megow (2022); Anand et al. (2022) and the references in the papers). Although we show that these measures perform fairly well, a huge drawback is the learnability and encoding length of information of these predictions. Thus, we also present an application-sensitive prediction model, which for most real-world instances is significantly smaller and easier to learn, yet still allows great consistency and robustness guarantees.

## PREDICTION MODELS

The two arguably most prominent prediction models in the literature are *full input* prediction and *action* prediction. In the former model, as the name states, the whole instance is predicted. This means that we are given the set of jobs along with their processing times and respective classes. In the latter, all actions are given, that is, a list of length $n$ with values between 1 to $m$ are given where the $t$th entry states on which machine to put job $j_t$. Predictions are either *static*, that is, we are given a list of $n$ machines and assign job $j$ to the $j$th machine in the list, or *adaptive*, meaning that upon each arrival of a new job $j$, we get the machine $i$ predicted on which $j$ should be placed on.

Both prediction models need to predict a significant amount of information because the encoding length of the information is dependent on the number of jobs $n$, and are hard to learn as they are sensitive to small changes in the instance. Thus, aiming for minimalistic predictions, we introduce a third prediction model specifically designed for this application named *class size* prediction. It predicts the total processing time of each class, but gives no information on the number of jobs nor specific processing times.

## OUR CONTRIBUTION

We overcome the daunting lower bounds for online class constraint scheduling by investigating the problem in a learning-augmented setting where an algorithm can access possibly erroneous predictions. We study three different prediction models: *full input* predictions, *action* predictions, and we introduce a new application-specific prediction model called *class size* prediction. For each of the prediction models, we design algorithms that beat the known lower bound of $m$ for online class constraint scheduling.

All prediction models lead to algorithms with different approximation guarantees and running times. As each of them can perform best for certain instances, and due to different eligibility of implementing such predictions, we believe that each of them is of independent interest. After presenting all results, we have a more in-depth evaluation of the different models.

Note that there is no purely online algorithm without access to predictions on the cardinality constraint scheduling which performs better than $m$ (see Theorem 1).

In this paper, we show the following algorithms, where $\ell$ is the error of the respective prediction: For minimizing the makespan for online class constraint scheduling on $m$ identical machines, there is

- an algorithm with predicted input with a competitive ratio at most $(\ell/OPT + \alpha)$ where $OPT$ is an offline optimal solution and $\alpha$ is the competitive ratio of an offline approximation algorithm (see Theorem 3).
- an algorithm with static predicted actions with a competitive ratio at most $(\ell + 1)$ (see Theorem 6).
- an algorithm running with class size predictions with a competitive ratio at most $\ell/OPT + 2 + \epsilon$ where $\epsilon$ is an accuracy parameter (see Theorem 11).

Input and action predictions, after carefully defining the model and error, yield, rather straightforwardly, algorithms with good consistency and robustness guarantees. However, the simplicity comes at the cost of large and difficult-to-achieve predictions. To minimize the information provided by the prediction, we thus consider the class size prediction model. Here, the information load is only dependent on the number of classes and not on the number of jobs.

For perfect class size predictions, we manage to improve the competitive ratio of online class constraint scheduling significantly from $m$ to a $2 + \epsilon$. To do so, we proceed in two steps: (i) we solve the predicted instance by treating each class as one splittable job and solve this problem using known results obtaining a schedule plan; (ii) then we post-process the schedule plan to obtain some exploitable structural properties. In particular, we define a structure that guarantees that even if the actual job sizes do not match the way we split the jobs in (i), we only get a bounded amount of extra processing time on each machine. This is achieved by defining a graph for the schedule plan, representing possible ways to move computation time from one machine to another without changing the classes present on each one. A cycle removal procedure is then applied to this graph to simplify it and introduce a hierarchy between the machines without changing the machines' individual makespans. This hierarchy is then used as a starting point to define how to place the incoming jobs and the unexpected loads that come with them.

In all prediction models, we assume that the number of classes is known to the algorithm. In fact, we prove that this is essential: if the number of classes is either not given or predicted possibly erroneously, then there exists no algorithm, regardless of which further (possibly erroneous) predictions are given that can beat a competitive ratio of $m$ (Theorem 1 and Corollary 2).

Finally, we prove all of our competitive algorithms to be (nearly) tight, i.e., no algorithm with access to the same predictions can yield a (significantly) better competitive ratio (see Theorems 4, 7, and 12).

FURTHER RELATED WORK

Packing and scheduling problems are widely studied, and empowering them with additional information has already been intensively studied. We recommend Chen & Potts (1998); Strusevich (2005); Christensen et al. (2016) for a survey on scheduling and packing algorithms and Dwibedy & Mohanty (2022) for a survey on semi-online scheduling. The task of minimizing the makespan on parallel identical machines is probably the most studied objective for scheduling since the sixties, and a considerable body of literature exists. It is well known to be NP-hard and admits a PTAS. Graham Graham (1966) is famous for his *List* scheduling algorithm for which he showed its $(2 - 1/m)$-competitiveness. It has been proven to be optimal for $m \in \{3, 4\}$ Faigle et al. (1989) and the competitive ratio has been successively lowered for higher number of machines to 1.923 Galambos & Woeginger (1993); Bartal et al. (1995); Karger et al. (1996); Albers (1999). The corresponding lower bound was also improved over the years to 1.85358 for $m > 80$ Gormley et al. (2000).

The more recent line of research on *learning-augmented* algorithms offers a novel approach that integrates additional (and possibly erroneous) predictions to the input to achieve better performances, if they are accurate, while maintaining good performance when they are not. These predictions are commonly divided into two categories: predicting a part of the online input (*input prediction*) Azar et al. (2022a;b); Bamas et al. (2020); Lykouris & Vassilvitskii (2018); Purohit et al. (2018), or pre-

dicting actions of the algorithm (*action prediction*) Antoniadis et al. (2020); Bamas et al. (2020); Lattanzi et al. (2020). The recent paper Lindermayr & Megow (2022) also proposes a more exotic type of prediction for non-clairvoyant scheduling (when the processing time of a job is not known until it is placed on a machine): *permutation prediction*, which can be seen as a sort of action prediction. In this input a model, a permutation of jobs is predicted that hints at a priority order. This builds on previous results that show that knowing the *Weighted Shortest Remaining Processing Time* order of jobs is enough to find an optimal schedule Smith (1956). Very recently, non-clairvoyant scheduling with precedence constraints (information about a job is revealed only when all of its predecessors have been completed) was also studied in the learning-augmented paradigm Lassota et al. (2023). To the best of our knowledge, none of these algorithms or ideas have been extended to work for the class constraint scheduling problem, so we fill this gap in this paper.

### STRUCTURE OF THIS DOCUMENT

We start by demonstrating that each individual prediction is incapable of surpassing the pessimistic bound of $m$ if the number of classes is not known beforehand or is subject to potentially erroneous predictions.

Following our goal to find minimalistic predictions for online class constraint scheduling, we investigate the prediction models by the amount of information they generate. That is, we first analyse full input predictions, then turn our attention to action predictions, and eventually arrive at the most involved and problem specific one: class size predictions. We conclude the paper with a discussion on the results and the relation between the prediction models.

## 2 KNOWING THE NUMBER OF CLASSES IS ESSENTIAL

In the following sections, we assume that the number of classes is always given and that this information is correct. We argue that these preconditions are necessary (but not sufficient) to design any algorithm that beats the trivial worst-case competitive ratio of $m$. This section is dedicated to proving this statement.

**Theorem 1.** *There exists no algorithm with a competitive ratio better than $m$ if the number of classes in the instance is not given (nor can be computed correctly from the given predictions).*

*Proof.* Let $m \geq 2$. Consider some machine $i$ that has a job of class $c$ assigned to it. No algorithm can decide to put any other job of class $c$ on any other machine than $i$ as the algorithm needs to guarantee the feasibility of any instance. Thus, it cannot choose to occupy extra class slots in case the instance has $mk$ many different classes.

Now assume an instance with $m$ equally sized jobs of the same class. Any algorithm has to stack these jobs on the same machine, where the optimal solution would have been to spread the $m$ jobs on the $m$ machines, yielding a competitive ratio of $m$. $\qquad\square$

The same argument holds if the number of classes is predicted, but might be erroneous as an algorithm needs to guarantee feasibility. Thus, we derive the following corollary.

**Corollary 2.** *There exists no algorithm with a competitive ratio better than $m$ if the number of classes in the input is (possibly erroneously) predicted.*

Note that this precondition is not enough on its own and does not prevent the lower bound from Epstein et al. (2022).

## 3 FULL INPUT PREDICTION

In this commonly assumed prediction setting, the whole input is predicted. That is, we get predicted all jobs along with their processing times, classes, and identifier to be able to map them to the actual arriving jobs. Processing times and classes can be predicted wrongly, but we assume the identifier to be correct and that each predicted class is among the set of actual classes in the instance, which is a natural restriction as commonly use cases are designed to work for a specific set of classes.

As proved in Section 2, knowing the correct number of classes present is necessary to overcome a competitive ratio of $m$.

**Full Input Prediction Algorithm**    The algorithm first computes a solution to the predicted instance. As the offline problem is already NP-hard, it can either spend exponential time to compute an optimal solution, or compute a near-optimal solution in polynomial time, e.g., using the PTAS by Jansen, Lassota, and Maack Jansen et al. (2020). We call this schedule $S$. Upon receiving job $j$, if the class was predicted correctly, the algorithm places the job as in $S$. Otherwise, job $j$ is placed on the machine with the least load in $S$ that has its class.

The prediction error $\ell$ is the sum of differences between the predicted processing time and actual processing time of a job $j$. Note that if the class was predicted wrongly, it can be interpreted as having a non-existent job predicted, thus its whole processing time is added to the error.

**Theorem 3.** *For minimizing the makespan for online class constraint scheduling on $m$ identical machines, there is an algorithm running in time $O(nmk)+T$ with predicted input with a competitive ratio at most $(\ell/OPT + \alpha)$ where $T$ is the time needed to compute the offline solution $S$ and $\alpha$ is the competitive ratio of an algorithm $A$ computing the offline solution $S$.*

*Proof.* If all predictions are correct, the Full Input Prediction Algorithm places each job as in the offline solution, yielding its competitive ratio $\alpha$. If a job is not predicted correctly, but the class was predicted correctly, it only adds its error to the makespan. Otherwise, its whole processing time might be added, which is captured by the error. Hence, the additive increase on the fullest machine is at most $\ell$.

Computing $S$ takes time $T$ per definition. Finding the corresponding job in $S$ for the incoming job takes time $O(n)$ as it might need to be compared to all jobs. If the class was predicted incorrectly, the lowest loaded eligible machine needs to be found in time $O(mk)$. This leads to an overall running time of $O(nmk) + T$. □

**Theorem 4.** *For minimizing the makespan for online class constraint scheduling on $m$ identical machines, no algorithm with static full input predictions can achieve a competitive ratio smaller than $(1 + \ell/OPT)$.*

In our proofs, we make use of an adversary that can manipulate the stream of jobs to incur the worst performance of the algorithm being studied. We consider the strongest form of adversary, namely an adaptive adversary that can react to the algorithm's decisions while it is running, as illustrated in the following proof:

*Proof.* Choose $k, \ell > 0$, $m \geq 2$ arbitrarily. Predict $mk$ jobs with processing time 1 and distinct classes. Any algorithm has to place $k$ jobs on each machine. This is clearly an optimal solution. The adversary now decides that all processing times of classes placed on machine 1 have in fact been predicted incorrectly. In particular, all $k$ classes on machine 1 get an additional job of $\ell/k$, yielding an increased makespan and an error of $\ell$. □

Theorem 3 and Theorem 4 together yield:

**Corollary 5.** *The Full Input Prediction Algorithm is tight w.r.t the competitive ratio.*

## 4   ACTION PREDICTION

In this variation, the actions (i.e. on which machine to place a job) are predicted. The error measure is the number of wrong predictions $\ell$. As no information is available on the upcoming jobs, the best strategy is to follow the predictions. In detail, we define the following algorithm.

**Action Prediction Algorithm**    Upon receiving a job $j$ and a predicted machine $P_j$, place $j$ on $P_j$ if possible (i.e. if there are still free class slots). Otherwise, put it on one of the eligible machines with the lowest load.

**Theorem 6.** *For minimizing the makespan for online class constraint scheduling on $m$ identical machines, there is an algorithm running in time $O(mnk)$ with static or dynamic predicted actions with a competitive ratio at most $(\ell + 1)$.*

*Proof.* If all predictions are correct, the Action Prediction algorithm obviously yields an optimal schedule.

When placing a job on a machine such that the machine gets a load bigger than $OPT$ then there might be two reasons: either the current prediction is wrong or there are no other possibilities, due to a previous wrong prediction.

If the current prediction is wrong, the additional error could reach the size of the largest job $p_{\max}$. We have $p_{\max} \leq OPT$. If the prediction could not be followed because that machine's class slots are already used, then there were previous wrong predictions that took this job's class slot. In the worst case, a previous error could result in moving the entire processing time scheduled on the original machine in the optimal schedule on top of another machine. Since $OPT$ is the maximum load on any machine in an optimal schedule, at most that much processing time can be moved to the current machine by one such previous error.

Thus, each error can only cause an increase of the makespan of at most $OPT$. Hence, $\ell$ errors can increase the makespan by at most $\ell \cdot OPT$.

Regarding running time, each step costs time at most $O(mk)$ to find and verify an eligible machine to place a job. In total, this yields a running time of $O(mnk)$. □

Note that $\ell$ can get as large as $n$, however, the maximal error is bounded by $m \cdot OPT$ by placing all jobs into the same machine.

**Theorem 7.** *For minimizing the makespan for online class constraint scheduling on $m$ identical machines, no algorithm with static or dynamic predicted actions can achieve a competitive ratio smaller than $\ell$ for $\ell \leq m - 1$.*

*Proof.* Let $k = 2$ and $m \geq 2$ be chosen arbitrarily. The instance contains $m + 1$ classes.

The adversary first produces a stream of jobs of size 1 and class 1. We predict machine $i$ for the $i$th job. This stream of jobs stops either when each machine gets exactly one job (following the prediction) or if two jobs get placed on the same machine.

In the first case, $m$ jobs have been produced. The adversary then produces $\ell$ large jobs of size $N >> 1$ and class 2. These jobs need to be stacked onto a single machine for the instance to stay feasible. The adversary then concludes with $m - 1$ jobs of size 1 and of classes 3, 4, ..., $m + 1$, which fill the remaining class slots. An optimal schedule would have placed the $m$ jobs of the first stream evenly on $m - \ell$ machines so that the $\ell$ large jobs can be spread among the remaining $\ell$ machines. Thus, we had $\ell$ errors in the prediction. $N$ is large enough for the makespan to basically be $\ell \cdot N$ instead of $OPT = N$ and we get a competitive ratio of $\ell$.

In the second case, we have stacked two jobs of size 1 on top of each other. The adversary then produces $m$ jobs of small size $\epsilon$ and classes 2, 3, ..., $m + 1$. An optimal solution would have spread the first $m$ jobs and placed the small jobs on top. There was no prediction error, and the makespan is $2 \cdot OPT$.

This shows that regardless of which decision we take, an algorithm cannot achieve a competitive ratio better than $\ell$. □

Theorem 6 and Theorem 7 together yield:

**Corollary 8.** *The Action Prediction Algorithm is tight with respect to the competitive ratio.*

## 5 CLASS SIZE PREDICTION

Next, we turn our attention to the application-sensitive prediction model: class size predictions. Here, instead of predicting the whole input, only the accumulated processing time of each class is

predicted. As before, the number of classes is assumed to be known. Intuitively, an error should capture the distance between the predicted classes and the actual ones. Indeed, we use this natural interpretation by using an $L_1$-type error measure: sum the absolute difference between the actual size of each class and its predicted size.

**Class Size Predictions Algorithm**    The algorithm proceeds in three steps:

1. Use an algorithm to solve the class/cardinality constraint scheduling problem on the predicted input (interpreting each class to be one splittable job) to obtain a schedule plan $S$.

2. Postprocess $S$ such that no two machines have the same subsets of classes for subsets larger than 1, obtaining the schedule $S'$. Define a hierarchy $H$ on the machines based on $S'$, which will be specified below.

3. Upon arrival of a job $j$ of class $c$, assign it to a machine using the order defined by $H$, where a machine can be used until it has no available processing time for class $c$ according to $S'$.

Roughly speaking, the algorithm follows a planned schedule to assign the arriving jobs. The main work lies in generating a planned schedule with the structural property that no two machines share the same subset of classes (for subsets larger than 1). Only then do our consistency and robustness guarantees hold. Note that, in contrast to full input predictions, we do not need to predict any job identifier. Further, we like to mention that one can use any algorithm to solve the class/cardinality constraint scheduling problem on the predicted input (interpreting each class to be one splittable job) to obtain a schedule plan $S$. Splittable means that a job does not have to be put on a machine as a whole, but instead, can be split arbitrarily into small pieces.

**First step**    We solve the class/cardinality constraint scheduling problem on the predicted input to obtain a preliminary schedule plan $S$. Note that we interpret every class to only contain one job, and that this job can be split arbitrarily.

Let $\alpha$ be the competitive ratio of the chosen algorithm $A$, and $T$ its running time. The best algorithm (running in polynomial time with respect to $n$) regarding the competitive ratio is an EPTAS provided by Chen et al. (2016). Of course, one could spend an exponential amount of time to compute an optimal solution, i.e., $\alpha = 1$.

**Second step**    Let $S$ be the schedule plan obtained in the first step. W.l.o.g., we assume that each class occupies a consecutive slot on a machine, otherwise the schedule can be rearranged without changing the makespan guarantee.

Let $G = (V, E)$ be a graph derived from $S$ as follows:

- generate two vertices $(x_1^{(i,c)}, x_2^{(i,c)})$ for each class $c$ present on each machine $i$ in $S$. We call the vertices $x_1^{(i,c)}$ *input* vertices, and $x_2^{(i,c)}$ *output* vertices, respectively,

- introduce an edge from each $x_1^{(i,c)}$ to each $x_2^{(i,c')}$ where $c \neq c'$, i.e., an edge from each input vertex on a machine $i$ to each output vertex of another class on the same machine $i$. We call these edges *conversion* edges, and

- insert an edge from each $x_2^{(i,c)}$ to each $x_1^{(i',c)}$ where $i \neq i'$, i.e. an edge from each output vertex to each input vertex of the same class on a different machine. We call these edges *transfer* edges.

We call this graph the *conversion and transfer* (CT) graph of $S$. See Figure 1 for an example.

We now aim to alter $S$ to satisfy our desired structural property. To do so, we introduce the following procedure which runs as long as there is a cycle $C$ in the CT-graph $G$ of $S$:

1. Identify the output vertex $x_2^{i,c}$ in $C$ corresponding to the class slot of smallest accumulated processing time $p_{min}$ in $S$.

2. For each transfer edge $(x_2^{j,d}, x_2^{j',d})$ in $C$, transfer $p_{min}$ processing time of class $d$ from machine $j$ to machine $j'$ in $S$.

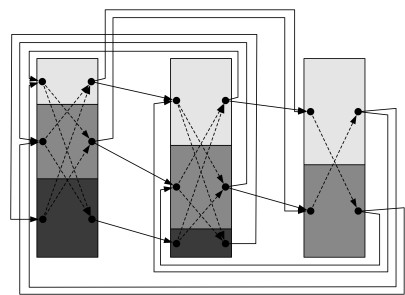

Figure 1: An example of conversion and transfer graph. Each column represents a machine and each rectangle a class slot. Input vertices are on the left of each class slot and output vertices are on the right. Conversion edges are dashed and transfer edges are continuous.

3. Delete all vertices corresponding to class slots with zero processing time on a machine (which holds at least for $x_1^{i,c}$ and $x_2^{i,c}$).

This procedure does not alter the makespan of the original schedule as for each machine along the cycle, there is as much processing time added as removed. Each iteration breaks a cycle, removes at least one class slot and never introduces more cycles or vertices, and thus terminates. Call the altered final graph $G'$ and the corresponding schedule $S'$.

**Third step**  We now use the acyclic structure of $G'$ to develop a strategy to assign jobs to machines. Let $\mathcal{M}(c)$ denote the set of all machines handling class $c$. Consider a hypergraph $H = (V, E)$ related to $G'$ with $V = \bigcup_{c \in \mathcal{C}} \mathcal{M}(c)$, i.e. the vertex sets correspond to all machines which admit at least one class in $S'$, and $E = \{(\mathcal{M}(c)), \forall c\}$, that is, each hyperedge connects for each class $c$ all machines that schedule class $c$ in $S'$. In the following, we prove that $H$ has a strong acyclic property, namely $H$ is Berge-acyclic (Berge (1985)).

**Definition 1.** *A bipartite incidence graph of a graph $G = (V, E)$ is the graph obtained by $G$ such that it has one vertex for each vertex and each edge in $G$, and an edge $(v, e)$, $v \in V, e \in E$, if $v \in e$ for $e \in E$ w.r.t. $G$.*

**Definition 2.** *A graph $G = (V, E)$ is Berge-acyclic if its bipartite incidence graph is acyclic.*

**Lemma 9.** *$H$ is Berge-acyclic.*

*Proof.* By definition $I$ is bipartite, and any cycle in it alternates between the two components of the bipartition. In a cycle $C$ each path of the form $v - e - v'$ exists if and only if $v, v' \in e$ and thus a transfer edge exists from $v$ to $v'$ and the other way around. Note that $v \neq v'$ because $C$ is a cycle and this directly reflects the fact that there exists no transfer edges from a machine to itself. Similarly, each path of the form $e - v - e'$ exists if and only if $e, e' \ni v$ and thus a conversion edge exists from $e$ to $e'$ and the other way around. Again, $e \neq e'$ because $C$ is a cycle and this is reflected in the fact that conversion edges only go to different classes inside any given machine.

Thus, there is a one to one correspondence between cycles in $G'$ and cycles in $I$. We know that $G'$ is acyclic and as a result $I$ is too, making $H$ Berge-acyclic. $\qquad \square$

Select a machine $i$ at will and set it to be the *root* of $H$. Since $H$ is Berge-acyclic, each hyperedge $c$ has a unique vertex $x_c$ that is closest to the root, i.e., reachable by only one hyperedge for classes not containing $i$, and $i$ itself with zero hyperedges for all other classes. The corresponding machine $j$ to $x_c$ is called the *parent* machine of class $c$. All other machines scheduling that class in $S'$ are *child* machines of class $c$, and cannot also be child machines of an other class. See Figure 2 for an illustration.

Given $H$ with the structural implications for $S'$, we can now state the rules to place an arriving job $j$ with class $c_j$:

1. if there is a child machine for $c_j$ that, according to $S'$, still has unassigned processing time for class $c_j$, place $j$ there,

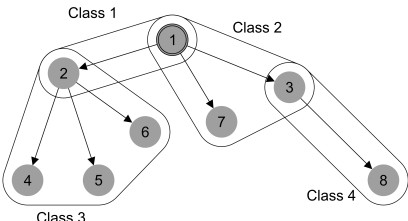

Figure 2: An example of hypergraph $H$. Class 1 is on machines 1 and 2, class 2 on machines 1, 3 and 7, class 3 on machines 2, 4, 5 and 6, and class 4 on machines 3 and 8. Machine 1 is the root and inside each class, arrows go from the parent machine to the child machines.

    2. else, place $j$ on the parent machine of $c_j$.

**Lemma 10.** *The job assignment rules above ensure that for any class $c$:*

    *1. the amount of excess in processing time of $c$ placed on a child machine of $c$ is at most $OPT$.*

    *2. the amount of excess in processing time of $c$ placed on the parent machine of $c$ never exceeds the prediction error for $c$.*

    *Proof.* Regarding the first property, a job $j$ is never placed on a child machine $i$ that has no available processing time for that class $c_j$. Further, $p_{max} \leq OPT$, thus if job $j$ exceeds the processing time assigned to $c_j$ on $i$, it can only do so by at most $OPT$. In addition to this, child machines can only be child machines of a single class. Indeed, if a machine has two different parent machines, then two different paths to the root exist in the incidence graph, which is prohibited by $H$ being Berge-acyclic. These two properties lead to the fact that overflows of different classes cannot accumulate on the same machine and that this overflow never exceeds $OPT$.

    Regarding the second property, a job is only placed on the parent machine $j$ when all the child machines are full; thus, the excess in processing time caused by jobs on the parent machine can only be smaller or equal to the prediction error for this class. □

The running time has to be analyzed separately for the preprocessing phase, which is run only once at the beginning, and the job assigning phase, which is run for each job. We first analyze the running time of the preprocessing performed before receiving any job. The cardinality constraint problem can be solved in almost linear time Chen et al. (2016):

$$2^{O(1/\varepsilon^2 \log^2(1/\varepsilon) \log\log(1/\varepsilon))} + \mathcal{O}(n \log n \log\log n)$$

The cycle removal procedure can be run in $O(m^2 k^2(m^2 + k^2))$. A cycle can be found in the incidence graph by running BFS in $O(|E| + |V|) = O(mk(k-1) + mk(m-1) + 2mk) = O(mk(m+k))$. Each iteration removes at least three edges (one conversion and two transfer), so in the worst case, the cycle removal runs in time $O(m^2 k^2(m^2 + k^2))$.

Assigning jobs is computationally cheap, as the algorithm only needs to find any child machine that is not overflowing if there is one or the parent machine if there are none. This can be computed by a single pass over all machines, running in $O(m)$.

Altogether, we get:

**Theorem 11.** *For minimizing the makespan for online class constraint scheduling on $m$ identical machines, there is an algorithm running in time $O(T + m^2 k^2(m^2 + k^2))$ with predictions of the processing time of each class with a competitive ratio at most $\ell/OPT + 1 + \alpha$, where $\alpha$ is the competitive ratio for an algorithm $A$ used to solve the offline class/cardinality constraint scheduling problem on the predicted instance, and $T$ is the running time of $A$.*

*Proof.* There are two types of overflow: the ones caused by bad placement due to jobs not summing up to the remaining time in a class slot and the ones due to class size prediction errors.

For the first kind, Lemma 10 states that overflows happen only on child machines and are of size at most $OPT$. Because a machine can only be a child machine of a single class, there cannot be an excess in processing time of more than $OPT$ on any machine, adding one to the competitive ratio $\alpha$ and leading to a competitive ratio of $1 + \alpha$.

For the second kind, in the worst case, the entire prediction error accumulates on a single machine adding $\ell/OPT$ to the competitive ratio.

Both types can happen on the same machine at the same time, thus we have a competitive ratio of $\ell/OPT + 1 + \alpha$. $\square$

We can show that this competitive ratio is optimal up to a constant factor:

**Theorem 12.** *For minimizing the makespan for online class constraint scheduling on $m$ identical machines, no algorithm with static class size predictions can achieve a competitive ratio smaller than $\ell/OPT$.*

*Proof.* Let $m \geq 2$, and $k \leq m$. The instance contains $mk$ classes. The adversary predicts that all classes are of negligible size $\epsilon$. We first get $mk$ jobs of distinct classes and negligible size $\epsilon$. Then, the adversary picks a machine $i$. For each class on $i$, the adversary sends a job of size $N \gg \epsilon$. This results in a prediction error $l = kN$ and a schedule with a makespan of $kN$ where the optimal solution would have been to put one big job per machine leading to an optimal makespan of $N$. The competitive ratio of this instance is thus $kN/N = k$ and $l/OPT = kN/N = k$ too. $\square$

Theorem 11 and Theorem 12 together yield:

**Corollary 13.** *The Class Size Prediction Algorithm is tight up to a constant factor with respect to the competitive ratio.*

## 6 CONCLUSION AND FUTURE WORK

We study online class constraint scheduling in a learning-augmented setting where an algorithm can access possibly erroneous predictions. We study different models of predictions, and thereby give a structured overview of what additional information helps in the design of better scheduling algorithms. In particular, we study full input, action, and the problem-specific class size predictions. For each model, we design an algorithm and analyse its consistency and robustness. Additionally, we prove for all algorithms that they are (nearly) tight, i.e., no algorithm can perform (significantly) better with respect to these measures.

The results show that no prediction model generally outperforms the others; their suitability heavily relies on the given circumstances. While all algorithms offer good robustness and consistency guarantees, the algorithm using action predictions is most efficient in terms of runtime. This efficiency comes from its direct prediction of actions, whereas the other models need to first compute such actions (i.e., the schedule plan) from the input data. This is NP-hard in general, but approximately optimal solutions can be derived by polynomial time approximation schemes. Access to accurate action predictions is the most challenging one, as small differences in the input might lead to very different optimal actions, which makes them hard to learn. Comparatively, input predictions are easier to learn. However, the amount of information is dependent on the number of jobs. Class size prediction in turn generates an amount of information just depending on the number of classes, which can be drastically less than the number of jobs making it the most achievable model to be learned.

For each of the models, we give (nearly) matching lower and upper bounds w.r.t. their consistency and robustness. This gives a quite complete picture of the prediction complexity of the problem, but there are interesting extensions to study. In particular, the assumption that we know the number of classes in advance could be relaxed if one allows for an error on the class slot limit. Further, we focused on identical machines, but extending the algorithms to also work on (un-)related machines would broaden the applicability to real-word settings.

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
