# OpenReview forum: "Minimalistic Predictions for Online Class Constraint Scheduling"
_ICLR.cc/2025/Conference — ICLR 2025 Poster_

### Official Review · Reviewer_3H61 · 2024-11-04

**Soundness:** 3
**Presentation:** 3
**Contribution:** 2
**Rating:** 5
**Confidence:** 3

**Summary:**

This paper studies online scheduling with class constraints. Here, the constraints impose that the number of distinct classes a machine can take is bounded. The offline version of the problem is known to be NP-hard while PTAS exists for it. For the online setting, as expected, there exist impossibility results for deducing competitive ratios. Given these hardness results, the paper considers scenarios where the scheduler has access to some information about the problem instance. Here, the information is given by some possibly noisy predictions about the problem instance. The paper studies three different models: (1) full input prediction, (2) action prediction, and (3) class size prediction. Algorithms that guarantee nearly tight competitive ratios for the settings are provided and analyzed in the paper.

**Strengths:**

1. Concrete problem settings with different prediction models are proposed and studied.
2. Tight algorithms for the three settings are provided with a rigorous theoretical analysis.

**Weaknesses:**

1. It is not clear whether the prediction models appear in practice.
2. No concrete practical application is discussed.

**Questions:**

Would it be possible to motivate the problem settings, especially the three prediction regimes, in the context of concrete practical applications? The reader would be interested in how the learning process and the prediction results can be obtained and how the prediction errors are estimated. Without discussing this, the paper would be a better fit for a theoretical computer science venue.

---

> ### Author Response · Authors · 2024-11-18
>
> Thank you for the review and your time. We appreciate the valuable feedback.
>
> Regarding the motivation for our problem setting, we are happy to elaborate on it. We will also include the discussion in the paper.
> The first two prediction models (input and action) represent classical approaches, see e.g.  (Purohit et al., 2018; Azar et al., 2021; 2022a; Im et al., 2021; Antoniadis et al., 2022; Bernardini et al., 2022; Erlebach et al., 2022; Bamas et al., 2020; Lindermayr & Megow, 2022; Eberle et al., 2022; Anand et al., 2022; Jin & Ma, 2022). They serve both as fundamental baselines for further model development and as realistic scenarios in many applications. For instance, input models are suitable when instances share common characteristics, as seen in recurring tasks like warehouse stock delivery. Similarly, action models are practical when prior solutions or AI-generated strategies are applied.
>
> The class size prediction model, however, is particularly interesting and distinct. It introduces a problem-specific model that needs less information, making it applicable to a broader range of scenarios. In many cases, the total duration of a task is known or constrained, though the specifics of task division may be uncertain (e.g., generating test results from doctors, lectures, or workforce allocation for stocking tasks). This setting naturally aligns with class constraints and approximate class (or task) size predictions, which can often be derived from historical data.

---

### Official Review · Reviewer_rcrR · 2024-11-04

**Soundness:** 3
**Presentation:** 2
**Contribution:** 3
**Rating:** 8
**Confidence:** 2

**Summary:**

The authors study the problem of online scheduling of tasks of possibly different classes and durations on $m$ machines, under the constraint that each machine can take care of at most $k$ different types of classes. More specifically, the authors study the problem of learning-augmented online scheduling, where predictions are given to the agent, and the objective is to design algorithms that achieve good performance if those predictions are accurate, while being robust if they are not.

The authors start by showing that no algorithm can achieve non-trivial regret when the number of classes is unknown. Then, they consider predictions on the class and durations of the jobs. They propose an algorithm, bound its competitive ratio and provide a lower bound for this problem, showing that this algorithm is optimal up to constant terms. Then, they consider predictions on the machine on which the jobs should be scheduled. Here again, they propose an algorithm, bound its competitive ratio and provide a lower bound for this problem, showing that this algorithm is optimal up to constant terms. Finally, they consider predictions on the total makespan of each class of jobs. They propose an algorithm, bound its competitive ratio and provide a lower bound for this problem, showing that this algorithm is optimal up to constant terms.

**Strengths:**

This papers adresses a natural problem, and study a variety of predictions types, providing a good understanding of this problem. The paper is mostly clearly written and self-contained. Although I am not an expert on the subject, and I can not really assess the novelty of the technics used, it appears to fill a gap in the set of existing results.

**Weaknesses:**

Some parts of the article could be rephrased to improve their clarity :
- In general, I believe it could be usefull to remind some definitions (objective, constraints, what is the optimal ratio etc) using mathematical formalism.
- I find the proof of Theorem 7 particularly unclear. Could you formalize more the problem instance you are considering?

Minor remarks :
- The beginning of Section 2 should read : "we assume", "we argue" instead of "we assumed", "we argued" (except if you are refering to the previous sections instead of the following sections?)
- The legend in Figure 2 does not seem to match the figure. Class 3 seems to be scheduled on machines 2,4,5,6 (instead of 3,4,5,6), and class 4 seems to be scheduled on machines 3,8 (instead of 3,4).

**Questions:**

Could you please consider rewriting the proof of Theorem 7?

In Section 5, it seems unclear to me why we can virtually allow to split jobs without paying for it in the competitive ratio. Could you discuss this fact?

---

> ### Author Response · Authors · 2024-11-18
>
> We thank the reviewer for their time and thoughtful feedback. The minor remarks were addressed and will show up in the next version. We will look into a clearer formulation of Theorem 7 and into recalling formal definitions.
>
> We would like to address the following points:
> "In Section 5, it seems unclear to me why we can virtually allow to split jobs without paying for it in the competitive ratio. Could you discuss this fact?".
>
> Answer:  we do actually pay for it in the competitive ratio as this is where the '+1' term comes from. In the worst case, a class is actually only a single, unsplittable job of size OPT.

---

> > ### Comment · Reviewer_rcrR · 2024-11-19
> >
> > I thank the authors for their answers.

---

### Official Review · Reviewer_QrRR · 2024-11-04

**Soundness:** 4
**Presentation:** 4
**Contribution:** 3
**Rating:** 8
**Confidence:** 3

**Summary:**

This paper studies online scheduling with class constraints. There are $m$ machines, $n$ jobs each with some processing time, jobs arrive one-by-one and must be placed on one of the $m$ machines. In addition to the standard goal of minimizing the makespan, there is a class constraint: each job belongs to one of finitely many classes and each machine can only support a total of $k$ classes. Since there are strong impossibility results in this setting, this paper considers the setting when the online algorithm has access to predictions about the input. The authors consider three different types of predictions and provide matching upper and lower bounds on the competitive ratios.

**Strengths:**

* The paper studies an important problem in online scheduling, i.e., that of class constraints.
* The paper studies the problem quite comprehensively - it shows the need for knowing the number of jobs, and proceeds to show upper and lower bounds on the competitive ratio for three different prediction models. Furthermore, the competitive ratios also show the dependence on the accuracy of the predictions.
* The algorithm for class size prediction is very interesting. For example, I liked how it leverages existing algorithms to come up with a schedule and then modifies it to arranges machines in a hierarchy.
* The paper is well written and easy to follow.

**Weaknesses:**

* Overall, this paper is well written. I am not *super* familiar with this specific sub-area, so I can't think of any major weaknesses. There are some minor presentation suggestions:
    1. I didn't understand Figure 1 until I read the text carefully. I suggest editing the figure to make it more self-contained. In its current form, the figure is very hard to understand.
    2. As I was reading the text and encountered the word "adversary", especially in your proofs, I wasn't super sure about the power of the adversary. It would be good to clarify whether it is oblivious or adaptive.
    3. Lines 103-107: It seems like this is formalized in Theorem 1? I found that easier to understand than this description. It might help to add a pointer to Theorem 1 here so that readers can refer to that instead if needed.

**Questions:**

1. Can you motivate the class size prediction model a bit more? Why does predicting the total processing time of each class make sense? Can you contextualize this with an application? (I suspect, for example, that one could use historical data and analyze the load of different types of jobs for a workflow and use that as the prediction. But it would be good to see a discussion of this in the paper.)
2. To compute the schedule plan, you assume that each class can be split into arbitrary-sized pieces. But the actual jobs themselves are not splittable - is this correct?
3. If you get more fine-grained information about the class sizes, do you think we could do better? For example, instead of the total processing time, maybe a distribution of the total processing time?

---

> ### Author Response · Authors · 2024-11-18
>
> We thank the reviewer for their time and thoughtful feedback. We will address the readability remarks in the coming days.
>
> Regarding the questions:
> 1. The class size prediction model makes sense for different reasons: as mentioned, it can be estimated with historical data and is thus practically relevant, but additionally, it reveals very little information about the instance, which was one of the goals of this paper. This means that the information provided by the predictions is small and easier to learn/approximate.
> 2. Yes, this is correct.
> 3. In general, more fine-grained information will probably lead to better performance by using more specific algorithms. That being said, if we get a distribution of the total processing time instead of the total processing time itself, we actually get less information about the instance and it will accordingly be more difficult to maintain a good performance. We could consider a distribution of job sizes for each class, but it is not clear that this would yield an improvement over our algorithms since the loss is mainly incurred by placing a class on the wrong machine (regardless of the job sizes). More specifically, the '+1' term in the class prediction model's competitive ratio could be lowered if we knew the size of the biggest job for instance.

---

> > ### Comment · Reviewer_QrRR · 2024-11-26
> >
> > Thanks for your response.

---

### Official Review · Reviewer_ZJ1K · 2024-11-10

**Soundness:** 3
**Presentation:** 3
**Contribution:** 1
**Rating:** 3
**Confidence:** 4

**Summary:**

The paper considers the problem of online makespan minimization with class constraints. There are $m$ machines and $n$ jobs, each of which have a class associated with them. The goal is to assign jobs to machines to minimize the makespan (the max load on any machine) subject to the constraint that no machines has more than $k$ distinct classes associated with it.
Although the offline problem admits an EPTAS, the online problem seems to be associated with strong lower bounds of $m$. In other words, a random assignment of classes to machines and then jobs to the classes is the best one can hope to do in this setting.

The authors consider this problem in the learning augmented setting where we are given a prediction about the instance before hand. The competitive ratio depends on the deviation from the true outcome from the prediction and the worst case competitive ratio.  The authors talk about three main types of prediction:

1) Full Input Prediction - The entire instance is predicted along with the class type and the process time.  The error is taken as the difference in the process times if the class prediction is correct and the entire process time if the class prediction is incorrect.
2) Full Action Prediction - The entire action is predicted of which jobs ...
3) Class Size Predictions - The total process time for each class is predicted but not hte number of jobs.

In all the models, the authors achieve competitive ratios better than $m$.

**Strengths:**

The algorithms they produce are fairly simple for the first two models. Essentially one simply computes the best according to the offline optimum and then adjusts for mistakes in a straightforward way using values.
The most interesting algorithm is for the class size prediction model where we only have a prediction about the total size of each class.  Here they show that one can compute the offline optimal for the whole sequence and then partition the set of used machines so that no two have the same set of classes on them. This can now define a hierarchy and a corresponding simple procedure shows how we can place the jobs as they arrive.

**Weaknesses:**

The contributions seems poor for a publication at ICLR. The only interesting algorithm is the for the class based model and even then the algorithm doesn't introduce any new techniques or ideas which generalize to other problems.

**Questions:**

N/A

---

### Meta-Review · Area_Chair_DL1J · 2024-12-22

**Metareview:**

This paper considers online scheduling in the "learning-augmented" setting with different types of predictions, and it studies how those predictions impact the competitive ratio. The main idea is that, without prediction, the online version problem is (often) impossible.

I had mixed feelings about this paper, as I find it quite interesting, even though the "learning-augmented" part is slightly misleading and this paper might be more adapted to a TCS community than to the average ICLR one.

But then, I figured out that there is a non-negligible intersection between those two, and this paper would be interesting for them, and might give new research direction to others.

**Additional Comments On Reviewer Discussion:**

We had concerned about the targeted community, but then we figured out that it would interest enough people.

---

### Decision · Program_Chairs · 2025-01-22

Accept (Poster)